# Graph-based Uncertainty Metrics for Long-form Language Model Outputs

**Mingjian Jiang**
Stanford University
jiangm@stanford.edu

**Yangjun Ruan**
Stanford University
ryoungj@stanford.edu

**Prasanna Sattigeri**
IBM Research
psattig@us.ibm.com

**Salim Roukos**
IBM Research
roukos@us.ibm.com

**Tatsunori Hashimoto**
Stanford University
thashim@stanford.edu

## Abstract

Recent advancements in Large Language Models (LLMs) have significantly improved text generation capabilities, but these systems are still known to hallucinate, and granular uncertainty estimation for long-form LLM generations remains challenging. In this work, we propose Graph Uncertainty – which represents the relationship between LLM generations and claims within them as a bipartite graph and estimates the claim-level uncertainty with a family of graph centrality metrics. Under this view, existing uncertainty estimation methods based on the concept of self-consistency can be viewed as using degree centrality as an uncertainty measure, and we show that more sophisticated alternatives such as closeness centrality provide consistent gains at claim-level uncertainty estimation. Moreover, we present uncertainty-aware decoding techniques that leverage both the graph structure and uncertainty estimates to improve the factuality of LLM generations by preserving only the most reliable claims. Compared to existing methods, our graph-based uncertainty metrics lead to an average of 6.8% relative gains on AUPRC across various long-form generation settings, and our end-to-end system provides consistent 2-4% gains in factuality over existing decoding techniques while significantly improving the informativeness of generated responses[1].

## 1 Introduction

Large Language Models (LLMs) [1–5] have demonstrated remarkable capabilities and been widely used as an interactive chatbot to provide knowledge and answers to user queries. However, they still struggle with generating false information, often referred to as "hallucinations" [6], which hinders their ability to provide calibrated [7–10] and factual [11, 12] responses and ultimately undermines the trust users place in their outputs. Improving uncertainty estimation techniques is crucial for building trust in LLMs and mitigating the risks associated with their deployment in real-world applications.

While many existing uncertainty estimation techniques for LLMs primarily focus on estimating the uncertainty of their answers to multiple-choice questions [13, 7, 8] or their entire generated responses (typically in a short form) [8, 14, 9, 15], they are often not sufficiently informative in real-world applications where LLMs generate paragraphs of texts consisting of a mixture of true and false claims [12]. In such scenarios, more granular uncertainty estimates are needed to help users distinguish the reliability of each individual claim within the generated text. Recent approaches [11, 16] attempt to measure the uncertainty of each claim by its consistency with randomly sampled responses based on the concept of self-consistency [17]. However, they do not fully leverage the semantic relationships between claims and responses that could support more granular uncertainty estimation.

---

[1]Our code is available at https://github.com/Mingjianjiang-1/Graph-based-Uncertainty.

38th Conference on Neural Information Processing Systems (NeurIPS 2024).

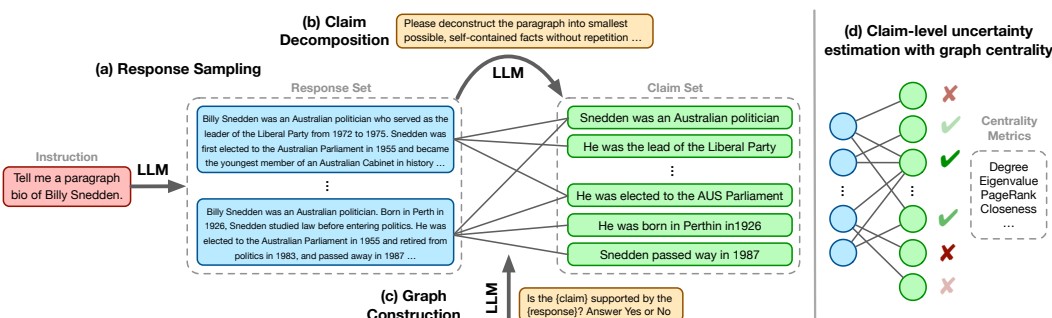

Figure 1: **Graph Uncertainty for claim-level uncertainty estimation**. We first sample several responses from LLMs (a) and decompose each response into atomic claims (b) following Sec. 4.1. The key components are the construction of a bipartite graph that captures the relations between responses and claims (c) and the use of graph centrality metrics to estimate the uncertainty of each claim. We simplify the pipeline and prompt for presentation, see Appx. F for details.

We introduce Graph Uncertainty (Fig. 1), a framework for granular, claim-level uncertainty estimation for LLMs which utilizes the fine-grained semantic relationships over a claim-response entailment graph. Our key idea is motivated by the observation that given multiple responses sampled from LLMs and a set of claims decomposed from them, we can construct a bipartite graph that captures the semantic entailment relationships between each response and claim, which serves as a summary statistic that captures uncertainty information associated with each response and claim. On this graph, claim-level uncertainties correspond to the *importance* of each node in the graph, and we extract such information with a family of graph centrality metrics [18–21] that measure the 'importance' of each claim node within the graph. A notable example is the existing uncertainty estimates based on the concept of self-consistency [17, 11], which turns out to be a special instantiation of our framework with the degree centrality as the uncertainty measure. Our framework generalizes it to accommodate a broader family of well-studied graph centrality metrics that capture more granular graph information than degree centrality. Through a systematic benchmarking of various graph centrality metrics and different baselines, we demonstrate that closeness centrality [18] is a natural and high-performance uncertainty estimator and outperforms baselines by an average of 6.8% on AUPRC at claim-wise uncertainty estimation for models like GPT-4 [2] on two challenging long-form generation datasets, FactScore [12] and PopQA [22].

Furthermore, we demonstrate that our granular, claim-level uncertainty estimates translate to gains in the factuality of LM outputs by integrating them with an uncertainty-aware decoding process that can leverage these graph metrics. The idea is straightforward and builds upon ideas explored in Mohri and Hashimoto [16], Wang et al. [23]: after obtaining claim-level uncertainty estimates following our method, we filter the claim set using uncertainty scores and synthesize the remaining claims with low uncertainty to produce the final response. Our empirical analysis demonstrates that our framework generates responses with better factuality without compromising their informativeness, achieving a better tradeoff compared to existing methods. Notably, our approach provides consistent 2-4% gains in factuality and can generate 70% more true claims at the 95% precision level compared to baselines.

The main contributions of our work are as follows:

- We introduce Graph Uncertainty, a general framework for claim-level uncertainty estimation of long-form LLM generations through the use of semantic graphs and graph centrality metrics.

- We demonstrate that our method significantly outperforms existing methods with an average of 6.8% gains on AUPRC for claim-level uncertainty estimation, and provide a systematic benchmarking of different baselines and our method with various graph centrality metrics in these settings.

- We present a straightforward and effective framework that integrates granular uncertainty estimates into LLM decoding for both factual and informative generations. Our method provides consistent 2-4% factuality gains and generates 70% more true claims than baselines at 95% precision level.

## 2   Related Work

**Short-form uncertainty estimation in LLMs** Existing approaches for characterizing the uncertainty of LLMs have largely focused on multiple-choice classification or short-form generation setups and

can be categorized into likelihood-based [13, 7, 24, 9], consistency/ensemble-based [10, 25], and verbalizer-based [14, 15] methods. In this line of work, Kadavath et al. [8] transforms uncertainty estimation into a binary classification task to estimate the probability of whether a sample is true or not. Kuhn et al. [9] proposes to estimate the 'semantic entropy' of the sample distribution given a prompt to address the surface-form competition [26] in the generated samples. Lin et al. [27] generalizes it to incorporate more fine-grained similarities between different samples and utilize degree or eigenvalue-related metrics for better discriminative performance. Xiong et al. [10] proposes a framework for systematically evaluating verbalizer and consistency strategies of eliciting confidence scores for black-box LLM, without access to model parameters or activations. Our work is distinct from these existing works in our focus on the long-form setting, which requires uncertainty estimation at a more granular level.

**Granular uncertainty estimation** Recent works have begun to extend uncertainty estimation to long-form outputs. Duan et al. [28] and Band et al. [29] obtain claim-level uncertainty scores from long-form outputs but operate in a white-box setting – requiring access to model internals – and therefore do not apply to API-based LLMs. Most relevant to our work, Manakul et al. [11] extends the concept of self-consistency [17] to assess uncertainty at the sentence level within long-form outputs, which is applicable to black-box LLM. Building upon this, Mohri and Hashimoto [16] performs uncertainty estimation at the claim level, combining a self-consistency style technique with conformal prediction. However, these works purely rely on the sample-and-count technique [11] that may not sufficiently capture the semantic relationships between claims and responses. Our work improves granular uncertainty estimation by considering more fine-grained semantic information contained in the entailment graph and its associated centrality measures.

**Factuality of LLMs** There have been several works on enhancing the factuality of LLMs at various stages, including retrieval [30, 31], pretraining [32, 33], and fine-tuning [34, 35, 29]. These methods typically require a reliable knowledge database for retrieval or extensive model training, which can be impractical and costly. Our work focuses on improving the factuality of LLMs at inference time and can be combined with other techniques. In this context, CoVe [36] proposes that LLMs can enhance their outputs through a series of planning and self-verification steps. DoLA [37] dynamically selects intermediate layers at each decoding step to minimize the generation of incorrect facts, though this method is only applicable to white-box LLMs. Recently, Wang et al. [23] and Mohri and Hashimoto [16] study methods that leverage claim-level uncertainty estimates for more factual LM outputs. We show that improved uncertainty estimators and decoders based on the entailment graph formalism lead to significant improvements in the factuality of LM outputs.

## 3   Preliminary

In this work, we focus on the problem of granular uncertainty estimation for LLMs, particularly in the context of distinguishing whether each claim in a long-form output is factual. Specifically, let $\Sigma$ denote the set of all characters and $\Sigma^*$ the space of all possible text strings. Given a text prompt $x \in \Sigma^*$, the generation process of a model $M$ with a specified temperature $T = t$ can be represented as a conditional probability distribution $M_{T=t}(\cdot|x)$ over $\Sigma^*$.

**Uncertainty estimation for LLMs** In the context of LLMs, uncertainty estimation is concerned with the following: given a model $M$, a prompt $x \in \Sigma^*$, and a response $y \in \Sigma^*$, we seek an uncertainty function $U : \Sigma^* \times \Sigma^* \to \mathbb{R}$ that measures the uncertainty of LLMs about the response. In this work, we define the efficacy of $U$ by how effectively it differentiates the true and false claims of $y$, using classification metrics such as AUROC and AUPRC. We focus on classification metrics rather than calibration or coverage, as our main goal will be using $U$ to identify and remove false claims as part of a decoding-time intervention.

**Claim-level uncertainty estimation** In many practical applications, the outputs from an LLM encompass a few paragraphs of text containing multiple *claims* [11, 16]. We consider a claim to be the smallest semantically distinct unit of information presented within the generated output. For example, in Fig. 1, "Snedden was elected to the Australian Parliament" is an example of a single claim. In this work, instead of assigning a single uncertainty score to the entire output, we assess uncertainty at the level of individual claims. Formally, we further define $\mathfrak{C}$ as a universal set of all unique, semantically distinct claims. The claim-level uncertainty function is then $U : \Sigma^* \times \mathfrak{C} \to \mathbb{R}$, allowing for granular analysis of factuality at the claim level.

Table 1: **Graph centrality metrics with their formulas and brief explanations.** We utilize the centrality metric value for each claim node to measure the uncertainty of the claim. Self-consistency-based estimate corresponds to the specific case of using the degree centrality. $V$ and $A$ are the node set and adjacency matrix of the graph $G$. A full definition of the notations is provided in Appx. E.

| Metric | Formula | Brief Explanation |
|---|---|---|
| **Degree** | $C_D(v) = \sum_{u \in V} A_{vu}$ | Number of edges incident to a node $v$ |
| **Betweenness** | $C_B(v) = \sum_{s \neq v \neq t} \frac{\sigma_{st}(v)}{\sigma_{st}}$ | Fraction of shortest paths $\sigma_{st}$ between other nodes $s, t$ that pass through a node $v$ |
| **Eigenvector** | $C_E(v) = \frac{1}{\lambda} \sum_{u \in N(v)} A_{vu} C_E(u)$ | Importance of a node $v$ measured by the importances of its neighboring nodes $N(v)$ |
| **PageRank** | $C_{PR}(v) = \frac{1-d}{|V|} + d \sum_{u \in N(v)} \frac{C_{PR}(u)}{N(u)}$ | Stationary distribution of a random walk within the graph with restart. $d$ is the damping factor. |
| **Closeness** | $C_C(v) = \frac{|V|-1}{\sum_{u \in V} d(v,u)} \cdot \frac{|V|}{|V_v|}$ | Reciprocal of the average shortest path distance to all nodes |

**Black-box LLM setup & Existing approaches** We focus on settings where we can only access the LLM outputs for each prompt and not its internal architecture or likelihood estimates. This reflects the real-world scenario for the most capable models, such as GPT-4 [2] and Claude-2 [3], which are typically accessed through API calls. The black-box assumption restricts the applicability of certain uncertainty estimation methods, such as those that rely on activation layers or logits [24, 37, 29]. There are relatively few methods that apply to this black-box setting. A few examples of uncertainty quantification methods that are applicable include:

- **Verbalized Confidence (VC)** [14, 15] based approaches which involve prompting the LLM to express its confidence in a claim $c \in \mathfrak{C}$ directly, based on the prompt $x \in \Sigma^*$. The uncertainty is quantified by parsing the verbalized confidence expression (e.g.,"very confident","100%", etc.) and mapping it to a numerical value.

- **Self-consistency (SC)** [17, 10, 11] based approaches involve checking the claim $c$ (typically decomposed from the greedily decoded output $M_{T=0}(x)$) against a set of sampled responses $\mathcal{R} = \{r^{(i)}\}_{i \in [N]}$ where $r^{(i)} \sim M_{T=t}(\cdot|x)$ at a higher temperature $t > 0$. The uncertainty estimate of $c$ is typically calculated by the proportion of responses $r^i$ that entail (denoted by $\Rightarrow$) the claim $c$, where $N$ is the total number of generated responses. Formally, the consistency score for a claim $c$ can be expressed as $\text{SC}(c) = \frac{1}{N} \sum_{i=1}^{N} \mathbb{1}[r^{(i)} \Rightarrow c]$. A higher consistency score indicates lower uncertainty, as the claim is more consistently entailed across the diverse responses.

Empirical evidence [10, 16] suggests that SC often surpasses VC in its effectiveness for uncertainty estimation.

## 4 Claim-Level Uncertainty Estimation with Semantic Entailment Graphs

Our motivation stems from the observation that given a set of generated responses $\mathcal{R}$ and their entailed claims $\mathcal{C}$, we can construct a bipartite graph $G = ((\mathcal{R}, \mathcal{C}), \mathcal{E})$ between $\mathcal{R}$ and $\mathcal{C}$ with edges $\mathcal{E}$ indicating the entailment relationship between each response and claim (Fig. 1). This graph captures the semantic entailment relationship between responses and claims, from which we may extract information that effectively captures the uncertainty of each claim. A motivating example is SC, which turns out to be a special case of calculating the degree centrality of each claim node as their uncertainty. This encourages the investigation of a broad family of graph-based metrics beyond the node degree, potentially offering more robust uncertainty estimates by exploiting intra-graph information.

In the following section, we will demonstrate how to construct the semantic entailment graph using an LLM in Sec. 4.1, and then describe the graph metrics we are exploring in Sec. 4.2.

### 4.1 Semantic Entailment Graph Construction

Here we describe the procedure for constructing a bipartite graph $G = ((\mathcal{R}, \mathcal{C}), \mathcal{E})$ that captures the generation-claim relationships for a given input $x$ using an LLM, as illustrated in Fig. 1. The graph construction involves multiple LLM interactions, with detailed prompts provided in Appx. F.

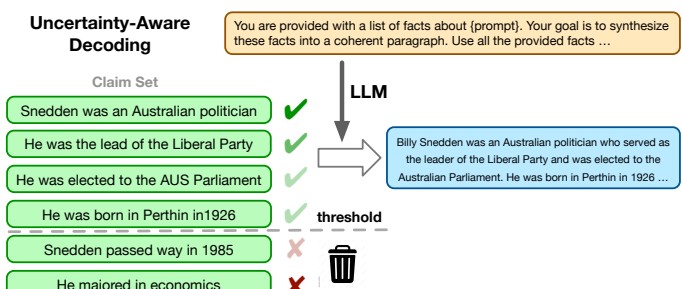

Figure 2: **Our uncertainty-aware decoding framework**. Based on our claim-wise uncertainty estimates obtained from Fig. 1, we keep low-uncertainty claims above a certain confidence threshold and use LLMs to synthesize them into a coherent response. Varying the threshold enables us to balance factuality and informativeness.

**Step 1: Response sampling ($\mathcal{R}$)** Given an input prompt $x$, we follow the same procedure as SC [11] to generate a set of $|\mathcal{R}|$ responses from the LLM, including one greedily decoded response $M_{T=0}(x)$ and $|\mathcal{R}| - 1$ responses from $M_{T=t}(\cdot|x)$. This process produces a set of generated responses $\mathcal{R} = \{M_{T=0}(x), M_{T=t}^{(1)}(x), \ldots, M_{T=t}^{(|\mathcal{R}|-1)}(x)\}$.

**Step 2: Claim decomposition and merging ($\mathcal{C}$)** We select a subset $\mathcal{R}_N \subset \mathcal{R}$ of $N$ responses from $\mathcal{R}$. For each response $r \in \mathcal{R}_N$, we first prompt the LLM to decompose $r$ into a set of claims, denoted as $\mathcal{C}_r$, following the procedure in [12]. Since the claims from different responses may semantically overlap, we also utilize an LLM to merge all claims into a comprehensive set of all unique, semantically distinct claims. Specifically, we prompt an LLM to implement a union function $\mathcal{M} : \mathcal{P}(\mathfrak{C}) \times \mathcal{P}(\mathfrak{C}) \to \mathcal{P}(\mathfrak{C})$, which takes two claim sets $\mathcal{C}^{(1)}, \mathcal{C}^{(2)}$ and merges them according to their semantic meaning. This is approximated by prompting the LLM to evaluate whether each claim in $\mathcal{C}^{(2)}$ is entailed by any claim in $\mathcal{C}^{(1)}$, and only those that are not are kept and appended to the original set. By sequentially prompting the LLM to merge the claim sets, we could get a union of all claims in $\mathcal{R}_N$, i.e., $\mathcal{C}^{(1)} = \mathcal{C}_{r_1}, \mathcal{C}^{(i)} = \mathcal{M}(\mathcal{C}^{(i-1)}, \mathcal{C}_{r_i})$ for $i \in \{2, \ldots, N\}$ where $r_i \in \mathcal{R}_N$. The final set forms our set of claim nodes $\mathcal{C} = \mathcal{C}^{(N)}$.

**Step 3: Edge construction ($\mathcal{E}$)** To construct the bipartite graph, we link the responses in $\mathcal{R}$ to the claims in $\mathcal{C}$. An edge $e$ between a response $r \in \mathcal{R}$ and a claim $c \in \mathcal{C}$ is established if $r$ entails $c$. The entailment relation is determined by prompting the same LLM $M$, following the procedure in [11].

### 4.2 Uncertainty Estimation with Graph Centrality Metrics

Recall that SC corresponds to using the specific claim node degree as the uncertainty estimate. Intuitively, the effectiveness of SC demonstrates that the more 'connected' a claim node is to other nodes in the graph, the more likely the claim to hold true. Drawing on this premise, we explore a broader family of graph centrality metrics [18–21] that measure the importance of a claim node within the graph from different angles, some of which may correlate with the factuality of the node to a greater extent. Specifically, denoting the graph $G = (V, A)$ here by its node set $V$ and adjacency matrix $A$, we assess the graph centrality metrics detailed in Table 1. These include the betweenness $C_B$, eigenvalue $C_E$, PageRank $C_{PR}$, and closeness centrality $C_C$. A full definition of the notations is provided in Appx. E. Note that we use the Wasserman and Faust (WF) improved formula [20] for closeness centrality to ensure applicability to disconnected graphs.

These centrality metrics are pre-defined to measure the importance of a node in a graph in different ways, and it is not clear a priori which types of centrality are useful for uncertainty estimation. Therefore, we carefully study all of these centrality metrics in various settings in our experiments (Sec. 6.1). We use these centrality metric values of the claim nodes within the bipartite graph as their confidence scores (i.e., the negative values as their uncertainty estimates), and evaluate the correlation between these metric values and the claim factualities. This analysis helps us identify the most empirically effective centrality metric for uncertainty estimation at the granularity of claims.

## 5 Uncertainty-Aware Decoding

We have introduced a graph-based technique for estimating the uncertainty at the level of individual claims. To demonstrate that our uncertainty estimation method translates to more factual LLM outputs, we now present a framework that integrates these uncertainty estimates at decoding time to improve the factuality of LLM outputs (Fig. 2). Similar to contemporaneous work on factuality-enhancing decoding [16, 23], we filter claims by uncertainty score to retain only confident claims. We show in our experiments that our use of the entire claim set (vs claims associated with a single output,

compared to other works) and our improved uncertainty metrics lead to improved factuality without compromising the informativeness of LLM outputs.

The intuition of our approach is to retain only the most confident claims and to synthesize them into a single coherent response. A detailed description of the steps is provided below.

1. **Create a candidate claim set with uncertainty estimates:** For a given prompt $x$, generate a candidate claim set $\mathcal{C}$ for the final output, along with their uncertainty estimates $U(x, c), \forall c \in \mathcal{C}$. Different approaches can apply here, and we follow the same procedure described in Sec. 4.2.

2. **Filter claims by uncertainty estimates**: Filter out claims from the candidate claim set with a high uncertainty estimate above a certain threshold $\delta \in \mathbb{R}$. This creates an operational subset of $\mathcal{C}$, $\mathcal{C}^o = \{c \in \mathcal{C} | U(x, c) < \delta\}$, containing claims with low uncertainties. The threshold $\delta$ can be selected either heuristically in an unsupervised manner or based on a percentile $q$ over training data claims, where the latter approach provides correctness guarantees with factuality probability levels determined by $q$ [16]. Following the supervised approach, we determine $\delta$ by selecting a percentile $q$ and computing the corresponding threshold on a small set of training data.

3. **Integrate selected claims:** Integrate the selected claims in the operational subset ($\mathcal{C}^o$) into a single, coherent output using an LLM. The prompt details for this step can be found in Appx. F.

In subsequent sections, we show that our approach provides favorable factuality-informativeness tradeoffs, where factuality is defined as the precision of the generated claims, and informativeness is defined as the number of true claims included in the output, analogous to the recall of true claims.

## 6  Experiments

In Sec. 6.1, we benchmark our proposed graph-based metrics and existing methods adapted for claim-wise uncertainty on two long-form factuality datasets, demonstrating the effectiveness of closeness centrality as a reliable uncertainty measure. In Sec. 6.2, we show that applying the closeness centrality metric with our uncertainty-aware decoding framework demonstrates the best informativeness-factuality trade-off for long-form generation and empirically analyze the impact of each component. Additionally, Sec. 6.3 presents an ablation study to investigate the factors contributing to the performance of closeness centrality and provide insights for interpretation.

### 6.1  Uncertainty Estimation

In this subsection, we empirically analyze different graph centrality metrics for uncertainty estimation and systematically benchmark existing methods adapted for claim-wise uncertainty estimation.

**Datasets and annotation** We evaluated the different uncertainty estimation methods on two challenging datasets, FActScore [12] and (long-form) PopQA [22], where even the most capable LLMs like GPT-4 [2] demonstrate frequent factuality failures. For each dataset, we randomly sampled 100 entities and generated a set of claims about each entity with their uncertainty estimates using our pipeline described in Sec. 4.1. This process yielded over 2000 claims on average for each evaluation setting. We briefly describe each dataset and their annotation details here, and include additional details in Appx. A:

- **FActScore** [12] is a widely used dataset for evaluating the factuality of long-form text generation for LLMs, containing entities sourced from Wikipedia. To assess the factuality of claims, we employed a similar pipeline to the one in their paper, classifying them as True, False, or Subjective using LLMs conditioned on the corresponding Wikipedia article. We specifically used GPT-4-Turbo due to its low classification error rate.

- **Long-form PopQA** [22] comprises of entities covering a diverse range of subjects. The original PopQA was not designed for long-form generation, we adapted it by adjusting the prompt to "Provide me with a paragraph detailing some facts related to `subject`".

  To ensure data quality, we filtered out entities that either lacked a Wikipedia page or had pages shorter than 1500 characters. The factuality of claims was evaluated by GPT-4-Turbo using the associated Wikipedia pages as reference, where longer Wikipedia pages were preferred as they typically provide more comprehensive coverage of entity information, thus reducing the risk of false negative annotations.

We also evaluated different methods on the **Natural Question** dataset [38] and observed consistent gains. Due to a higher rate of false negatives in the auto-annotation pipeline compared to the aforementioned datasets, we have included these results in Appx. C.1.

Table 2: **Our claim-level uncertainty estimate based on the closeness centrality metric consistently and significantly outperforms baselines.** We assess the AUROC (ROC) and AUPRC-Negative (PRC) to compare baselines and different centrality metrics (Table 1) with the number of samples $|\mathcal{R}| \in \{5, 10\}$ on the FactScore and PopQA datasets. Results with statistically significant gains are bolded. All p-values are significantly less than 0.05 through a pairwise significance test detailed in Appx. C.2. Each setup is annotated as "model, $|\mathcal{R}|$". Abbreviations are used for baselines, as defined in the baseline discussion, and for centrality metrics, as defined in Sec. 4.2.

| | Setup Metric | GPT-3.5, 5 ROC | PRC | GPT-3.5, 10 ROC | PRC | GPT-4, 5 ROC | PRC | GPT-4, 10 ROC | PRC | Llama-3, 5 ROC | PRC | Llama-3, 10 ROC | PRC |
|---|---|---|---|---|---|---|---|---|---|---|---|---|---|
| FactScore | IL-VC | 0.537 | 0.437 | 0.537 | 0.437 | 0.540 | 0.394 | 0.540 | 0.394 | 0.536 | 0.486 | 0.536 | 0.486 |
| | PH-VC | 0.748 | 0.641 | 0.748 | 0.641 | 0.701 | 0.531 | 0.701 | 0.531 | 0.675 | 0.593 | 0.675 | 0.593 |
| | P(True) | 0.609 | 0.521 | 0.609 | 0.521 | 0.756 | 0.633 | 0.756 | 0.633 | 0.580 | 0.500 | 0.580 | 0.500 |
| | SC | 0.835 | 0.726 | 0.852 | 0.756 | 0.812 | 0.651 | 0.839 | 0.708 | 0.801 | 0.712 | 0.829 | 0.761 |
| | SC+VC | 0.870 | 0.781 | 0.879 | 0.801 | 0.827 | 0.670 | 0.838 | 0.701 | 0.817 | 0.739 | 0.833 | 0.771 |
| | $C_B$ | 0.765 | 0.706 | 0.793 | 0.731 | 0.758 | 0.637 | 0.780 | 0.683 | 0.743 | 0.695 | 0.759 | 0.733 |
| | $C_E$ | 0.794 | 0.726 | 0.809 | 0.735 | 0.775 | 0.623 | 0.797 | 0.673 | 0.732 | 0.690 | 0.757 | 0.722 |
| | $C_{PR}$ | 0.852 | 0.776 | 0.853 | 0.777 | 0.791 | 0.644 | 0.801 | 0.675 | 0.757 | 0.683 | 0.764 | 0.700 |
| | $C_C$ | **0.892** | **0.848** | **0.898** | **0.859** | **0.843** | **0.733** | **0.855** | **0.751** | **0.857** | **0.837** | **0.867** | **0.850** |
| PopQA | IL-VC | 0.508 | 0.449 | 0.508 | 0.449 | 0.499 | 0.325 | 0.499 | 0.325 | 0.568 | 0.473 | 0.568 | 0.473 |
| | PH-VC | 0.623 | 0.533 | 0.623 | 0.533 | 0.640 | 0.415 | 0.640 | 0.415 | 0.663 | 0.556 | 0.663 | 0.556 |
| | P(True) | 0.614 | 0.656 | 0.614 | 0.656 | 0.642 | 0.585 | 0.642 | 0.585 | 0.572 | 0.583 | 0.572 | 0.583 |
| | SC | 0.758 | 0.676 | 0.789 | 0.721 | 0.744 | 0.516 | 0.771 | 0.572 | 0.735 | 0.616 | 0.756 | 0.652 |
| | SC+VC | 0.778 | 0.693 | 0.794 | 0.716 | 0.752 | 0.548 | 0.762 | 0.581 | 0.761 | 0.666 | 0.775 | 0.692 |
| | $C_B$ | 0.650 | 0.645 | 0.683 | 0.679 | 0.718 | 0.514 | 0.738 | 0.566 | 0.702 | 0.607 | 0.720 | 0.638 |
| | $C_E$ | 0.716 | 0.666 | 0.728 | 0.697 | 0.703 | 0.564 | 0.729 | 0.604 | 0.700 | 0.591 | 0.725 | 0.630 |
| | $C_{PR}$ | 0.734 | 0.687 | 0.740 | 0.713 | 0.703 | 0.473 | 0.723 | 0.511 | 0.718 | 0.617 | 0.734 | 0.645 |
| | $C_C$ | **0.792** | **0.754** | **0.809** | **0.774** | **0.786** | **0.668** | **0.800** | **0.693** | **0.772** | **0.699** | **0.784** | **0.713** |

**Baseline methods** Since existing approaches mostly focus on uncertainty estimation at the generation level, we adapted them to claim-level estimation for the purpose of baseline comparison. In particular, we conducted a systematic benchmarking of a wide range of methods including:

- **Post-hoc verbalized confidence (PH-VC)** [8] This method is a variant of **Verbalized confidence (VC)** as introduced in Sec. 3, which elicits the verbalized confidence in a post-hoc manner after the entire claim set $\mathcal{C}$ has been decomposed from generations, following Tian et al. [15].

- **In-line verbalized confidence (IL-VC)** [16] This method directly elicits the verbalized confidence about each claim $c$ in an in-line manner right after it is decomposed from the generations during Step 2 in Sec. 4.1 and incurs negligible inference overhead in contrast to PH-VC.

- **P(True)** [8]: this method elicits the uncertainty estimate of a claim by prompting an LLM to answer whether the claim is true or false, using the likelihood of being true as the confidence score (which we estimated by sampling multiple times from the LLMs).

- **Self-Consistency (SC)** [17, 11]: as discussed in Sec. 3, this method utilizes the consistency score of one claim across different samples from the same LLM.

- **Self-Consistency + PH-VC (SC + VC)**: a straightforward variant of SC that integrates the PH-VC by summing their scores to break the frequent ties in confidence scores in SC.

For our graph-based uncertainty estimates, we evaluated four graph centrality metrics: **betweenness** ($C_B$), **eigenvalue** ($C_E$), **PageRank** ($C_{PR}$), and **closeness** ($C_C$). We provide more detailed information about our methods and baselines in Appx. B.

**Evaluation metrics** We assess how well various uncertainty estimation metrics distinguish factual claims from false ones using the Area Under the Receiver Operating Characteristic (AUROC) curve. Since the dataset may be imbalanced, we also compute the Area Under the Precision-Recall Curve for the Negative class (AUPRC-Negative). AUPRC-Negative focuses on the classifier's performance in identifying false claims that are more critical to the factuality compared to true claims. By using AUPRC-Negative, we can better assess the metrics' performance in identifying false claims, which complements the overall performance assessment provided by the AUROC metric. More detailed results are provided in Appx. C.

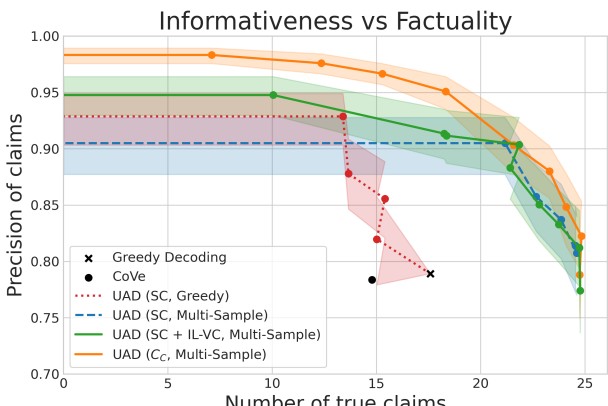

Figure 3: **UAD with better claim-level uncertainty estimates demonstrates a better trade-off between factuality and informativeness of the generated responses.** We compare UAD across different thresholds $\delta$ and two non-uncertainty decoding baselines. We assume that random noise is applied to break ties for each uncertainty method, resulting in a horizontal line extending from the leftmost dot to the left. The shaded confidence interval is obtained by bootstrapping with a confidence level of 95%.

**Experimental details** Our experiments are conducted on the three most capable LLMs to date (as of June 2024): GPT-3.5-turbo, GPT-4 [2], and Llama-3-70B-Instruct [39]. We used the same LLM to sample responses and construct a semantic entailment graph for uncertainty estimation as described in Sec. 4.1. The graph construction is set up as follows: To construct the set of claims $\mathcal{C}$, we used a greedily decoded sample (temperature $t = 0$) and 4 samples with temperature $t = 1$ as $\mathcal{R}_N$. To construct the set of responses $\mathcal{R}$ in the graph, we used $|\mathcal{R}| = 5$ or $|\mathcal{R}| = 10$ samples, where we included those for obtaining the claims and sampled additional ones with temperature $t = 1$ if needed. The impact of these hyperparameters is analyzed in Sec. 6.3. We collected all the claims in the entities we sampled and labeled their factuality using the method discussed in Sec. 6.1. Then, we only used those claims that are annotated as True or False (but not Subjective) to avoid ambiguity. The results are presented in Table 2.

**Closeness centrality consistently and significantly outperforms baselines** As illustrated in Table 2, our closeness centrality metric consistently outperforms other graph centrality metrics and baselines, including those with higher inference costs, such as SC + VC. In some settings, closeness centrality achieves significant gains up to over 6% at AUROC and 12% at AUPRC-Negative compared to the SC baseline. To better understand the effectiveness of closeness centrality for uncertainty estimation, we provide an ablation study in Sec. 6.3.

**Additional analysis** Our systematic benchmarking in Table 2 also provides some insights into the effectiveness of baseline methods for claim-level uncertainty estimation:

- **SC is a strong baseline**: We find that Self-Consistency (SC) score, which is essentially degree centrality $C_D$, serves as a strong baseline, outperforming all other previous approaches. It sometimes underperforms our PageRank centrality metric $C_{PR}$, but the comparison is sensitive to the specific setup or dataset.
- **IL-VC usually underperforms PH-VC**: Comparing the first two rows of each setup, we find that combining the process of claim decomposition and uncertainty elicitation of claims actually hurts the uncertainty estimation performance.
- **Integrating VC improves SC:** We observe that integrating VC into SC can improve its uncertainty estimation in most setups, even when a naive addition of their scores (SC + VC) is applied. This is probably because VC offers additional information to distinguish claims with tied scores in SC.

### 6.2 Uncertainty-Aware Decoding

In this subsection, we empirically demonstrate how our improved claim-level uncertainty estimates contribute to a better tradeoff between precision (factuality) and recall (informativeness) of generated responses within our UAD framework, and analyze the impact of each component.

**Experimental setup** We tested UAD with our best closeness centrality uncertainty estimate $C_C$ on the FActScore dataset used in Sec. 6.1. We experimented with GPT-3.5-Turbo that were used for all steps described in Fig. 1 and Fig. 2. We randomly sampled 180 entities and used $|\mathcal{R}| = 5$ to construct candidate claim set $\mathcal{C}$. Additional details can be found in Appx. B.2.

**Evaluated methods** We benchmarked the performance of UAD against several inference-time decoding methods. For a systematic comparison, we also included inference-time decoding methods that do

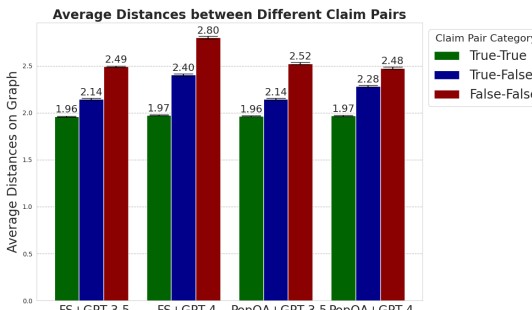
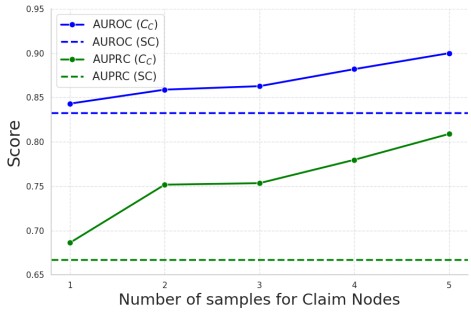

(a) Average length of shortest paths between claim pairs

(b) Effect of the size of $\mathcal{C}$ measured by $|\mathcal{R}_N|$

Figure 4: **Ablation study**: (a) The false claims have a greater average distance to other claims compared to true ones, indicating the effectiveness of the closeness centrality metric. (b) Performance improves consistently as we increase the number of responses $|\mathcal{R}_N|$ used to construct the claim node set $\mathcal{C}$ in our uncertainty estimation method. While all evaluations are conducted on the same fixed set of claims, varying $|\mathcal{R}_N|$ alters the graph structure used to estimate these claims' uncertainty values.

not rely on uncertainty estimation. For UAD, we evaluated its performance across various uncertainty estimation techniques $U$ and claim candidate set choices $\mathcal{C}$ to study their impact. Specifically, our evaluated methods include (Appx. B) :

- **Greedy Decoding:** the most naive baseline that generates a response with a temperature of $t = 0$ for a given input prompt $x$.

- **CoVe [36]**: an inference-time decoding method that improves the factuality of generations by self-verification without any uncertainty estimates being used.

- **UAD (SC, Greedy)**: as discussed in Sec. 5, this method corresponds to Conformal Factuality Decoding [16], which employs the SC uncertainty estimate to filter out high-uncertainty claims in the claim set obtained from the greedily decoded response.

- **UAD (SC, Multi-Sample)**: based on the previous method, it expands the claim set $\mathcal{C}$ to include those decomposed from multiple sampled responses $\mathcal{R}$, following our procedure in Fig. 1. Comparing this method with the previous one studies the impact of candidate claim set size $|\mathcal{C}|$.

- **UAD (SC + IL-VC, Multi-Sample)**: similar to the previous method, but it utilizes IL-VC to break ties for SC scores. We applied IL-VC instead of PH-VC here to ensure a fair comparison with similar inference costs across different methods (additional results with PH-VC are included in Appx. D.1).

- **UAD ($C_C$, Multi-Sample)**: it applies our best graph-based uncertainty estimate with the closeness centrality $C_C$ as $U$, from which we can study how better claim-wise uncertainty estimates may improve the performance of UAD.

**Evaluation Metrics** The efficacy of long-form text generation was assessed along two dimensions: factuality and informativeness of the generated content. The factuality score, ranging from 0 to 1, was measured using FactScore without length penalty [12], with higher being better. The informativeness score quantified the number of claims within the output. In cases where no claims were included in the output (e.g., "I don't know"), the factuality score is set to 1 but the informativeness score is set to 0, indicating the absence of potentially hallucinated content. These metrics were averaged across data, showing both the truthfulness and utility aspects of the generated text.

**Results and analysis** Our results are presented in Fig. 3, with factuality on the y-axis and informativeness on the x-axis. Methods positioned towards the upper right are preferred, as they demonstrate desirable performance for both factuality and informativeness. UAD-based methods trace a trajectory in the plot when varying the uncertainty estimation threshold $\delta$, while Greedy decoding and CoVe appear as single points. The results reveal several key findings:

- **UAD achieves a better tradeoff between factuality and informativeness:** We observe that UAD-based methods consistently outperform non-UAD methods, with all UAD variants achieving superior factuality-informativeness tradeoffs compared to Greedy Decoding and CoVe. Specifically, when generating the same number of claims as Greedy Decoding, UAD methods achieve up to

18% higher factuality. This indicates the effectiveness of utilizing uncertainty estimates to steer the response generation.

- **Expanding the claim candidate set $\mathcal{C}$ trades-off factuality for informativeness**: Comparing UAD (SC, Multi-Sample) and UAD (SC, Greedy), we find that by expanding $\mathcal{C}$ to include those decomposed from multiple samples, we can achieve a better trade-off in setups where more claims are desired, but a lower peak accuracy otherwise. This indicates that the choice of $\mathcal{C}$ should be determined based on the desired balance between factuality and informativeness.

- **Better claim-level uncertainty estimates lead to a better tradeoff**: Comparing UAD with SC, SC + IL-VC, and $C_C$ in Fig. 3, we find that applying our best metric, closeness centrality, also leads to a clearly dominating Pareto optimality. Our approach consistently achieves 2-4% gains in factuality and can generate 70% more true claims at the 95% precision level compared to the best baseline. Moreover, because closeness centrality provides a more fine-grained metric than degree centrality (used in SC), it offers a broader range of trade-off options compared to SC (even when combined with IL-VC to break ties), as evidenced by more points in the figure obtained by varying the threshold $\delta$.

## 6.3 Ablation Study

In this section, we aim to analyze why closeness centrality is effective for discriminating between true and false claims, and how the performance of this method changes as we increase the number of claim nodes in the semantic bipartite graph. For all experiments in the ablation study, we used the FActScore dataset with the GPT-3.5-turbo model.

**Why is the closeness centrality so effective?** Closeness centrality is intrinsically related to the distances between nodes in a graph. To understand its effectiveness, we analyze how the distances between a pair of claims correlate with the factuality of these claims. Specifically, in Fig. 4a, we visualize the distances between true-true claim pairs, true-false pairs, and false-false pairs in the semantic graph. We observe a clear shift in the distance distribution from true claims to false claims across all settings. False claims generally exhibit larger distances to other claims in the semantic entailment graph. This observation can be intuitively interpreted as false claims being less centered, i.e., less entailed by generations and having lower co-occurrences with the majority of claims. This insight provides a clear explanation for the strong performance of closeness centrality in claim-level uncertainty estimation.

**How does the performance change with the number of responses?** Fig. 4b explores how the performance changes as we increase the number of responses $\mathcal{R}_N$ used to construct the claim set. We find that increasing the number of claim nodes consistently leads to improved performance (despite the increased inference cost). This also indicates that the closeness centrality effectively leverages additional graph information as more nodes are present in the semantic graph.

**Is our end-to-end decoding pipeline computational intensive?** While our pipeline increases the lower bound of computation by introducing the graph construction process, it attains Pareto optimality for the compute-quality tradeoff compared to existing methods across multiple quality metrics. Details are present in Appx. D.2.

## 7 Conclusion

In this work, we propose Graph Uncertainty, a family of graph-based methods for claim-wise uncertainty estimation in LLM generations. We also present an uncertainty-aware decoding framework that integrates these estimates to improve the trade-off between factuality and informativeness. Empirical results demonstrate the effectiveness of the proposed approach.

Despite these improvements, we note that the graph construction process increases inference-time compute and latency. Moreover, our claim decomposition assumes that claims can be decontextualized and separated, which may not always be the case in real-world applications. Future work may aim to optimize the graph construction process to reduce computational overhead and develop techniques to handle scenarios with dependent claims more effectively. These improvements will further improve the applicability and robustness of our uncertainty estimation framework in complex settings.

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

# A   Data and Annotation Details

**FActScore**   FActScore [12] is a widely used dataset for evaluating the factuality of long-form text generation, containing entities sourced from Wikipedia. We utilized the entities from this dataset and applied our pipeline, as discussed in Sec. 4, to generate, break down, and evaluate the uncertainty of claims. To assess the factuality of sub-claims, we employed a similar pipeline, classifying them as True, False, or Subjective using GPT-4-Turbo, which was chosen for its low error rate. We used the 'unlabelled' split of FActScore released dataset.

**Long-form PopQA Dataset**   We also incorporated the PopQA dataset [22], which comprises entities covering a diverse range of subjects. Although PopQA was not originally designed for long-form generation, it contains challenging entities and provides links to corresponding Wikipedia pages. We filtered the dataset based on the length of the Wikipedia pages and the quality of the results. The factuality of claims was evaluated using the information from the associated Wikipedia pages, with longer pages considered more reliable due to their comprehensive coverage of the entity.

**Annotation Process**   For both datasets, we employed a two-stage annotation process. First, we used our pipeline to generate claims for each sampled entity. Second, we assessed the factuality of the generated claims using the following criteria:

- True: The claim is factually accurate and supported by the information provided in the corresponding Wikipedia page.

- False: The claim is factually incorrect or contradicts the information provided in the corresponding Wikipedia page.

- Subjective: The claim is subjective, opinion-based, or cannot be verified using the information provided in the corresponding Wikipedia page.

To ensure the quality of the annotations, we utilized GPT-4-Turbo, a large language model known for its low error rate, to classify the claims into the three categories mentioned above. This automated annotation process allowed us to efficiently label a large number of claims while maintaining a high level of accuracy.

# B   Baselines Details

- **Verbalized confidence (VC)**: As introduced in Sec. 3, this method involves prompting the LLM to express its confidence in a claim $c$ directly. Here, we mainly consider two variants:

  - **Post-hoc verbalized confidence (PH-VC)** [15]: This method elicits the verbalized confidence in a post-hoc manner after the entire claim set $\mathcal{C}$ has been decomposed from generations. We followed Tian et al. [15] to prompt an LLM to express its confidence about each claim $c \in \mathcal{C}$ given multiple options such as "Very good chance (80%)", "Little chance (20%)" etc.
  - **In-line verbalized confidence (IL-VC)** [16]: This is a straightforward variant of PH-VC, which directly elicits the verbalized confidence about each claim $c$ in an in-line manner right after it is decomposed from the generations during the Step 2 in Sec. 4.1. Notably, in contrast to PH-VC which requires an extra stage of prompting for VC, this method adds negligible overhead to the claim decomposition stage.

- **P(True)** [8]: This method elicits the uncertainty estimate of a claim by prompting an LLM to answer whether the claim is true or false, using the likelihood of being true as the confidence score. Since the likelihood scores are not available for most close-source black-box LLMs, we modified it to estimate the likelihoods by sampling multiple times from the LLMs.

- **Self-Consistency (SC)** [17, 11]: As discussed in Sec. 3, this methods utilizes the consistency score of one claim across different samples from the same LLM.

- **Self-Consistency + PH-VC (SC + VC)**: Since there are often tied confidence scores in SC (i.e., the frequencies of being it entailed in the sampled generations), rendering them less informative for distinguishing true and false claims at a granular level. We thus include a straightforward variant of SC by integrating the PH-VC to break such ties, which serves as a strong baseline in our experiments.

## B.1 Uncertainty Estimation Experiments

This section aims to provide comprehensive details about the baseline methods used in Sec. 6.1. We will delve into the specifics of each method, including their prompts and implementation, and any preprocessing or postprocessing steps applied. By offering this detailed explanation at these baselines, we aim to ensure the reproducibility and transparency of our experimental setup.

Assume that we have a claim $c$ to estimate uncertainty and and an entity of the claim.

**Post-hoc Verbalized Confidence (PH-VC)**   This method elicits the verbalized confidence in a post-hoc manner after the entire claim set $\mathcal{C}$ has been decomposed from generations. We followed Tian et al. [15] to prompt an LLM to express its confidence about the claim $c$ given multiple options such as "Very good chance (80%)", "Little chance (20%)" etc. The specific prompt that we use is as following:

```
You are provided with some possible information about a person.
Describe how likely it is that the possible answer is correct as one
of the following expressions:
No chance (0%)
Little chance (20%)
Less than even (40%)
Fairly possible (60%)
Very good chance (80%)
Almost certain (100%)
Give ONLY your confidence phrase, no other words or explanation.  For
example:
Confidence:  <description of confidence, without any extra commentary
whatsoever; just a short phrase!>
The entity is:  {entity}
The possible information is:  {claim}
```

**In-line Verbalized Confidence (IL-VC)**   The In-line Verbalized Confidence (IL-VC) method differs from PH-VC in terms of prompting and integration with claim decomposition. Thus, we prompt language model with a long-form generation and instructions to give all the claims with corresponding confidence scores. The specific prompt that we adapted from [16] is as following:

```
Please deconstruct the following paragraph into the smallest possible
standalone self-contained facts without semantic repetition, and
return the output as a jsonl, where each line is claim:[CLAIM],
gpt-confidence:[CONF].
The confidence score [CONF] should represent your confidence in
the claim, where a 1 is obvious facts and results like 'The earth
is round' and '1+1=2'.  A 0 is for claims that are very obscure or
difficult for anyone to know, like the birthdays of non-notable people.
The input is:
{long-form generation}
```

**P(True)**   The P(True) method estimates the uncertainty of a claim by prompting an LLM to answer whether the claim is true or false and using the likelihood of being true as the confidence score. Since the likelihood scores are not available for most closed-source black-box LLMs, we modified the method to estimate the likelihoods by prompting the model 10 times and frequency of answering True. The specific prompt that we adapted from [8] is as follows:

```
The following claim is about entity.  Is the claim true or false?
(Answer with only one word True/False)
Claim:  {claim}
```

**Self-Consistency (SC)**    The Self-Consistency (SC) method utilizes the consistency score of one claim across different samples from the same LLM. As we discussed in Sec. 3, it is equivalent to the degree of claim on the bipartite semantic entailment graph. The whole pipeline used to construct the graph is described detailed in Sec. 5.

**Self-Consistency + PH-VC (SC + VC)**    The combination of SC and PH-VC is simply the average of the two scores.

**Graph Metrics (Our Method)**    All the graph metrics are calculated by calling corresponding centrality function in networkx package.

### B.2    Uncertainty-Aware Decoding

We also provide a detailed explanation of the uncertainty-aware decoding methods used in our experiments.

**Greedy Decoding**    Greedy decoding is the most naive baseline that generates a response with a temperature of $t = 0$ for a given input prompt $x$. This widely acknowledged method produces outputs with high likelihood but does not incorporate any uncertainty estimates.

**CoVe [36]**    CoVe is an inference-time decoding method that improves the factuality of generations by self-verification without using any uncertainty estimates. We utilize the code released in this repo: https://github.com/ritun16/chain-of-verification.

**UAD Methods**

As the pipeline of UAD described in Sec. 5 involves three steps: Generate claim pool, estimate uncertainty and filter, and merge the claims into a coherent long-form generation. We use the same prompt present in Appx. F for integrating all claims into one sample for all methods.

Thus, for each method below, I will talk about the other two steps in details:

**UAD (SC, Greedy)**    As discussed in Sec. 5, this method corresponds to Conformal Factuality Decoding [16], which employs the SC uncertainty estimate to filter out high-uncertainty claims in the claim set obtained from the greedily decoded response. The claim pool is obtained by breaking down the greedy output $M_{t=0}(x)$, utilizing the break down prompt in Appx. F, and we use SC detailed in Appx. B.1 as uncertainty estimation method.

**UAD (SC, Multiple-Sample), UAD (IL-SC, Multiple-Sample), UAD ($C_C$, Multiple-Sample)**    Similar to the UAD (SC, Greedy) setting, the claim pool is obtained by using $|\mathcal{R}| = 5$ to construct candidate claim set $\mathcal{C}$ as discussed in Sec. 4.1. Then, we use respectively use SC, $C_C$, SC + IL-VC detailed in Appx. B.1 as uncertainty estimation method.

### B.3    Computing Resources

In this work, only experiments that are using Llama-3 involved computing resources. We use two 80G A100 to run inference for Llama-3-70B-Instruct.

## C    Additional Results for Uncertainty Quantification

We present additional experimental results and analyses related to uncertainty estimation. These experiments complement the main experiments discussed in Sec. 6.1 and provide further insights into the performance of different uncertainty estimation methods.

### C.1    Natural Question Dataset

To further evaluate the generalizability of our claim-level uncertainty estimation method, we expanded our experiments to include the Natural Questions dataset [38]. Results are presented in Table 3.

Our findings show that the closeness centrality method continues to outperform baseline approaches in most scenarios, aligning with trends observed in our primary datasets. However, we noted a higher rate of false negatives in the auto-annotation process for this dataset compared to others. While

Table 3: **Additional Results on Natural Question Datasets:** Our claim-level uncertainty estimation method outperforms baseline approaches across most scenarios, maintaining comparable results in others, with a single exception noted.

| | Setup | GPT-3.5, 5 | | GPT-3.5, 10 | | GPT-4, 5 | | GPT-4, 10 | | Llama-3, 5 | | Llama-3, 10 | |
| | Metric | ROC | PRC | ROC | PRC | ROC | PRC | ROC | PRC | ROC | PRC | ROC | PRC |
|---|---|---|---|---|---|---|---|---|---|---|---|---|---|
| NaturalQ | IL-VC | 0.536 | 0.228 | 0.536 | 0.228 | 0.674 | 0.318 | 0.674 | 0.318 | 0.539 | 0.252 | 0.539 | 0.252 |
| | PH-VC | 0.536 | 0.228 | 0.536 | 0.228 | 0.452 | 0.176 | 0.452 | 0.176 | 0.465 | 0.206 | 0.465 | 0.206 |
| | SC | **0.674** | 0.352 | 0.683 | 0.380 | 0.731 | 0.384 | 0.751 | 0.412 | 0.737 | 0.387 | 0.735 | 0.353 |
| | SC+VC | 0.670 | 0.341 | 0.677 | 0.356 | 0.596 | 0.280 | 0.607 | 0.293 | 0.634 | 0.295 | 0.633 | 0.268 |
| | $C_C$ | 0.666 | **0.38** | 0.682 | **0.408** | **0.734** | **0.413** | 0.752 | **0.439** | 0.736 | **0.432** | **0.746** | 0.403 |

this may slightly reduce confidence in these specific results, we believe they remain valuable in demonstrating the potential broader applicability of our method.

These results underscore the robustness of our approach across diverse question-answering contexts, while also highlighting areas for potential refinement in auto-evaluation pipelines for future work.

### C.2    Statistical Significance Check

To rigorously evaluate the performance differences between our proposed uncertainty estimation method and baseline methods, we conduct a statistical significance check focusing on the two metrics that is reported in Sec. 6.1, AUROC and AUPRC-Negative.

We use bootstrapping to generate multiple samples from the original dataset by resampling with replacement. For each bootstrap sample, we calculate the each metric for both the proposed and baseline methods. This results in a distribution of the metric values for each method.

Our goal is to validate the effectiveness of our best proposed method. To compare these distributions, we employ the Wilcoxon signed-rank test, a non-parametric test suitable for paired samples. A statistically significant result, indicated by a p-value less than 0.05, would confirm that the performance improvement of our method is meaningful and consistent.

After the significance check, all p-values for the three models (GPT-3.5-turbo, GPT-4, and Llama-3-70B-instruct) and two datasets (FactScore and PopQA) are significantly smaller than 0.05 when comparing the proposed method against the baseline methods.

### C.3    AUROC Plots

To provide a visual comparison of the performance of different uncertainty estimation methods, we present AUROC plots for selected experimental settings. These plots illustrate how our proposed method outperforms the baselines in distinguishing between true and false claims.

Figure 5a shows the AUROC curves for the GPT-3.5 model with $|\mathcal{R}| = 10$ on the FActScore dataset. We compare our best-performing graph-based uncertainty estimation method using closeness centrality ($C_C$) with the baselines, including post-hoc verbalized confidence (PH-VC), self-consistency (SC), and self-consistency combined with verbalized confidence (SC + VC). The plot clearly demonstrates that our method achieves a higher AUROC than all the baselines (and all points are higher), indicating it is more capable to identify false claims.

Figure 5b presents the AUROC curves for the GPT-4 model with $|\mathcal{R}| = 10$ on the PopQA dataset. Again, we observe that our method using closeness centrality ($C_C$) achieves a higher overall AUROC compared to the baselines. Interestingly, while closeness centrality consistently outperforms the SC baseline, the SC + VC method exhibits better performance in the left region of the plot, where the False Positive Rate (FPR) is low. This finding further validates the observation that incorporating VC can potentially enhance the performance of graph-based methods in certain scenarios.

## D    Additional Results for Uncertainty-Aware Decoding

### D.1    Additional Results on the Plot

In this section, we present additional results for the Uncertainty-Aware Decoding (UAD) experiments, where we include the variant using PH-VC (Post-Hoc Verbalized Confidence) to break ties for the

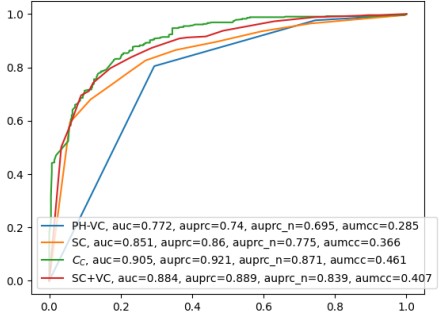
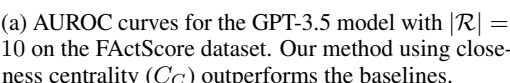

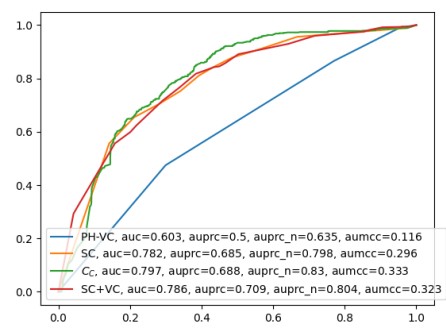

(a) AUROC curves for the GPT-3.5 model with $|\mathcal{R}| = 10$ on the FActScore dataset. Our method using closeness centrality ($C_C$) outperforms the baselines.

(b) AUROC curves for the GPT-3.5-turbo model with $|\mathcal{R}| = 10$ on the PopQA dataset.

SC (Self-Consistency) scores. While the main experiments in Section 6.2 focused on using IL-VC (In-Line Verbalized Confidence) to ensure a fair comparison with similar inference costs across different methods, here we provide the results using PH-VC for completeness.

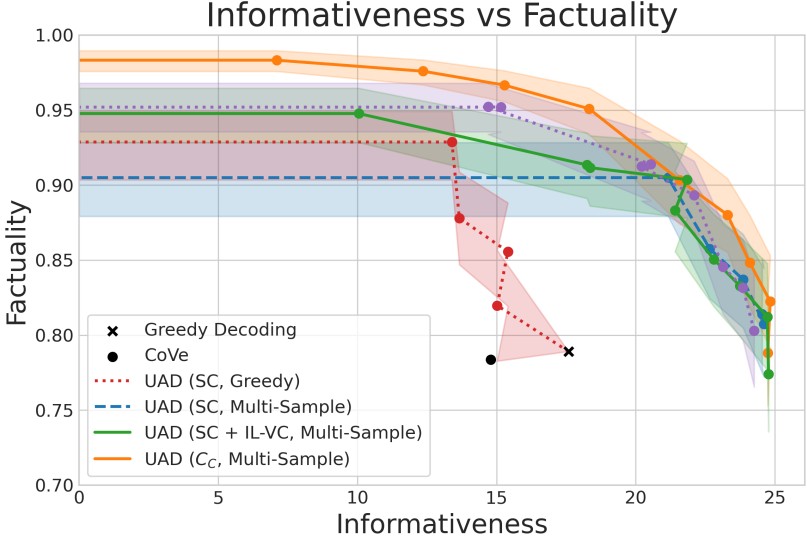

Figure 6: UAD results with PH-VC included for breaking ties in the SC scores. The plot shows the trade-off between factuality and informativeness for various UAD variants and baselines.

Figure 6 presents the updated plot, which includes the additional UAD variant:

- **UAD (SC + PH-VC, Multi-Sample)**: This method is similar to UAD (SC + IL-VC, Multi-Sample), but it uses PH-VC instead of IL-VC to break ties for the SC scores. As Sec. 6.1 shows, PH-VC achieves better performance on the two datasets than IL-VC.

The results show that UAD (SC + PH-VC, Multi-Sample) achieves a better trade-off between factuality and informativeness compared to UAD (SC + IL-VC, Multi-Sample), especially in the region where more claims are desired. This suggests that using post-hoc verbalized confidence estimates can indeed help to improve the performance of UAD, albeit at the cost of additional inference computation.

However, it is important to note that our best graph-based uncertainty estimate, UAD ($C_C$, Multi-Sample), which still dominates the tradeoff between the baselines. This highlights the effectiveness

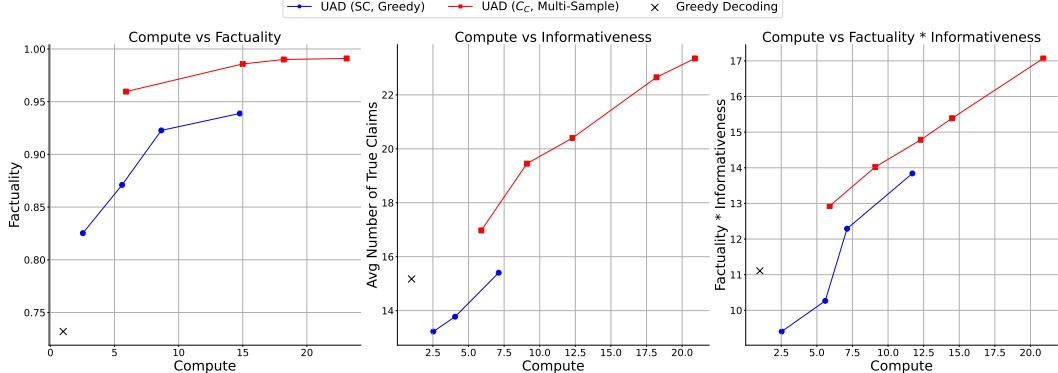

Figure 7: **Computational Cost Analysis of End-to-end Decoding Pipelines**. Each plot represents the best performance achieved after hyperparameter tuning at various compute budgets. The x-axis in subplots (a), (b), and (c) represents the computational cost as a multiple of greedy decoding. The y-axes show factuality (accuracy), informativeness (average number of true claims), and their product (a heuristic trade-off metric), respectively. Our method demonstrates Pareto optimality compared to greedy decoding and UAD (SC, Greed) baselines, indicating superior computational efficiency in achieving comparable improvements across different metrics.

of our proposed closeness centrality-based uncertainty estimation method in steering the response generation towards more factual and informative outputs.

### D.2   Computation Analysis

To evaluate the effectiveness and efficiency of our proposed method in improving the quality of LLM's outputs, we conducted an analysis comparing our approach to existing techniques. Our primary goal was to assess the trade-off between computational cost and output quality, focusing on factuality and informativeness. In our analysis, we compared compute costs and quality metrics (factuality, informativeness, and their product as a heuristic trade-off metric) against established baselines such as (SC, Greedy) and greedy decoding. Notably, we excluded baselines with comparable computational costs due to similar graph computation procedures. This decision was based on our previous findings, presented in Figure 3, which demonstrated that these baselines are outperformed by our method. Since a given compute budget might have different hyperparameter choices, we performed extensive hyperparameter tuning for both our method and the (SC, Greedy) baseline, plotting frontier curves for various configurations.

The results of our analysis, illustrated in Fig. 7, demonstrate that our method consistently outperforms baseline approaches across all metrics, achieving Pareto optimality in the compute-quality trade-off. Ultimately, while our method introduce a higher lower bound of computation, our results underscore the computational efficiency of our approach in achieving comparable improvements across different quality metrics, compared to previous black-box decoding methods.

## E   Notation Definition

In this section, we provide the definitions of the notations used in Sec. 4.2 for clarity and completeness. Let $G = (V, A)$ denote a graph, where $V$ is the set of nodes and $A$ is the adjacency matrix. The elements of the adjacency matrix $A_{vu}$ indicate the presence of an edge between nodes $v$ and $u$.

The following notations are used in the formulas for the graph centrality metrics in Table 1:

- $v$: A node in the graph $G$.
- $u$: Another node in the graph $G$.
- $s, t$: Source and target nodes, respectively, when considering shortest paths.
- $\sigma_{st}$: The number of shortest paths between nodes $s$ and $t$.
- $\sigma_{st}(v)$: The number of shortest paths between nodes $s$ and $t$ that pass through node $v$.
- $N(v)$: The set of neighboring nodes of node $v$.

- $\lambda$: The largest eigenvalue of the adjacency matrix $A$.
- $d$: The damping factor in the PageRank algorithm, typically set to 0.85.
- $d(v, u)$: The shortest path distance between nodes $v$ and $u$.
- $|V|$: The total number of nodes in the graph.
- $|V_v|$: The number of nodes in the connected component containing node $v$.

These notations are used consistently throughout the methodology section to define and explain the various graph centrality metrics employed for uncertainty estimation of claims in the bipartite graph representation.

# F    Prompt Details

In this section, we provide the specific prompts used in our methodology (Sec. 4, Sec. 5) for constructing the semantic entailment graph and performing uncertainty-aware decoding. These prompts are designed to elicit the desired information and behavior from the language model.

## F.1    Claim Decomposition Prompt

Given a LM response, the claim decomposition prompt is used to break down the generated response into a set of individual claims. The prompt is the same as the In-Line VC prompt because it decomposes and elicit confidence at the same time. The prompt is as follows:

```
Please deconstruct the following paragraph into the smallest possible
standalone self-contained facts without semantic repetition, and
return the output as a jsonl, where each line is claim:[CLAIM],
gpt-confidence:[CONF].
The confidence score [CONF] should represent your confidence in
the claim, where a 1 is obvious facts and results like 'The earth
is round' and '1+1=2'. A 0 is for claims that are very obscure or
difficult for anyone to know, like the birthdays of non-notable people.
The input is:
{long-form generation}
```

This prompt is applied to each generated response in the set $\mathcal{R}$ to obtain the corresponding set of claims $\mathcal{C}r$ for each response $r$.

## F.2    Claim Merging Prompt

Given two sets of claims, the claim merging prompt is used to combine two sets of claims into a single set which is a union set, ensuring that only unique and semantically distinct claims are retained. The prompt is as follows:

```
Given two lists titled "Original Claim List" and "New Claim List",
your task is to integrate information from the "New Claim List" into
the "Original Claim List". Please follow these detailed steps to
ensure accuracy and clarity in the process:
Task 1.  **Verification Process:** Your goal is to go through each
statement in the "New Claim List" one by one, and determine if it is
fully entailed or mentioned by any statement in the "Original Claim
List."
Task 2.  **Compilation of Non-Entailed Claims:** Generate a list of
statements from the "New Claim List" that are not already covered or
implied by the "Original Claim List." For each new or unique claim
that does not have an equivalent in the original list, format your
output by starting each line with a dash ('-').
**Original Claim List:**
{claim_list1}
**New Claim List:**
{claim_list2}
```

```
Begin with the Verification Process to assess each claim's relevance
and uniqueness, followed by the Compilation of Non-Entailed Claims to
clearly list any new insights that the "New Claim List" provides.
```

This prompt is used to sequentially merge the claim sets for each response $r_i \in \mathcal{R}$ to accumulatively obtain the final set of claim nodes $\mathcal{C}$.

### F.3 Edge Construction Prompt

The edge construction prompt is used to determine whether a generated response entails a specific claim, thereby establishing an edge between the response node and the claim node in the bipartite graph. We adapt the prompt from [11]. The prompt is as follows:

```
Context: {generation}
Claim: {claim}
Is the claim supported by the context above?
Answer Yes or No:
```

This prompt is applied to each pair of response $r \in \mathcal{R}$ and claim $c \in \mathcal{C}$ to construct the edge set $\mathcal{E}$ of the bipartite graph.

### F.4 Claim Integration Prompt

The claim integration prompt is used to synthesize the selected low-uncertainty claims into a single, coherent output during the uncertainty-aware decoding process, as mentioned in Sec. 5. The prompt is as follows:

```
Task:  You are provided with a list of facts about prompt.  Your
goal is to synthesize these facts into a coherent paragraph.  Use
all the provided facts where possible, ensuring that no fact is
misrepresented or overlooked.  If there are redundant facts, choose
the most comprehensive one for inclusion.  The length of the paragraph
should naturally reflect the number of provided facts-shorter for
fewer facts and longer for more.  Avoid unnecessary filler and focus
on presenting the information clearly and concisely.
The facts:
{claim_list}
```

This prompt is applied to the operational subset of claims $\mathcal{C}^o$ obtained after filtering based on the uncertainty threshold $\delta$ to generate the final output $M(\mathcal{C}^o)$.

These prompts play a crucial role in the construction of the semantic entailment graph and the implementation of uncertainty-aware decoding. By using these prompts, we can effectively leverage the language model's capabilities to extract claims, establish relationships between responses and claims, and generate coherent outputs based on the selected low-uncertainty claims.

