# OpenReview forum: "Graph-based Uncertainty Metrics for Long-form Language Model Generations"
_NeurIPS.cc/2024/Conference — NeurIPS 2024 spotlight_

### Official Review · Reviewer_2gkm · 2024-07-09

**Soundness:** 4
**Presentation:** 4
**Contribution:** 3
**Rating:** 7
**Confidence:** 3

**Summary:**

In this work, the authors design graph-based uncertainty metrics for long-form generation. Specifically, they sample multiple responses to a query and decompose them into claims. They then create edges between these responses and claims based on their entailment relationships. Using the constructed graph, they found that closeness centrality is the most effective metric for measuring the uncertainty of claims. Additionally, they propose uncertainty-aware decoding (UAG), which enhances the LM’s factuality by filtering out low-confidence claims and reconstructing them into a coherent response.

**Strengths:**

Originality: The idea of introducing graph metrics to estimate claim uncertainty is novel. The authors also design a new decoding paradigm for long-form generation.

Quality: Experiments indicate the effectiveness of the proposed methods. The evaluation is thorough, and the analysis is concise and to the point.

Clarity: The paper is well-written and easy to follow.

Significance: The introduced uncertainty estimation method helps users assess the reliability of generated content. The decoding method also contributes to generating more accurate content.

**Weaknesses:**

1. Both the uncertainty estimation method and UAG require multiple samples from the LLM, which is time-consuming and might significantly increase computational cost.

**Questions:**

1. Just want to double check, is C_D equal to Self-consistency?

2. As mentioned in the weaknesses, computational efficiency might be an issue for your method. I suggest the authors add a discussion on this. At least report the additional cost compared with the vanilla LM and the API cost before and after applying your methods.

3. Based on my understanding, in the responses from the datasets used in this paper, there are fewer dependencies between claims. Do you consider other long-form generation tasks where there are more complex inter-sentence dependencies? For example, a QA pair in the ELI5 style:

Query: What’s the purpose of statute of limitations?

Response: Because evidence that might prove the defendant's innocence is lost with time. For example, ask me what I was doing a month ago, and I can probably find something like a bank statement, etc. to show I was at XYZ bar or something. Ask me for something from 2010? That's harder to track down.

This is an example supporting the previous claim, “evidence that might prove the defendant’s innocence is lost with time.”

4. The following related work is missing:

[1] LitCab: Lightweight Language Model Calibration over Short- and Long-form Responses

**Limitations:**

A clear limitation of this work is computational efficiency, which restricts its practical application. Addressing this issue might not be trivial. Another suggestion for future work is that the authors could conduct experiments about how these estimated uncertainties benefit users when they are prompting LLMs.

---

> ### Author Rebuttal · Authors · 2024-08-07
>
> Thank you for your valuable feedback and suggestions. We appreciate that you acknowledged our method “novel” and our evaluation “thorough”. We would like to address your remaining questions and concerns in the following response.
>
>
> ### Computational cost
>
> > *”Both the uncertainty estimation method and UAG require multiple samples from the LLM, which is time-consuming and might significantly increase computational cost.”*
>
> > *”As mentioned in the weaknesses, computational efficiency might be an issue for your method. I suggest the authors add a discussion on this. At least report the additional cost compared with the vanilla LM and the API cost before and after applying your methods.”*
>
> We agree that the computation overhead and time latency is the major limitation of our method. We'll address these concerns by discussing them respectively.
>
> **Computation cost**  While our method does increase the lower bound of computation, we would like to emphasize that our method offers a desirable option for utilizing more inference-time computation to improve the quality (i.e., factuality & informativeness) of LLMs’ responses. Our approach provides various hyperparameters that allow for flexible adjustment of the performance-cost trade-off.
>
> In response to your suggestion, we conducted experiments to explore the computational efficiency required to achieve equivalent quality improvements in output. Our new results demonstrate that our method achieves **Pareto optimality in the compute-quality trade-off** compared to existing methods.
>
> **Pareto optimality** Our analysis compares the computational costs and different quality metrics (factuality, informativeness, and their product as a combined metric) for our method against the (SC, Greedy) baseline and greedy decoding. We performed an extensive evaluation by tuning hyperparameters (generation count, claim quantity, and threshold settings) for both our approach and the (SC, Greedy) baseline, then mapped out the efficiency frontiers. We excluded certain baselines with comparable computational demands due to similar graph processing methods, such as (SC + IL-VC, Multi-Sample), and (SC, Multi-Sample), as Figure 3 clearly shows our method's superiority over these. The comprehensive findings are presented in Figure 5 of the supplementary material. Therefore, while the increase of computation cost is noticeable, our method could be an desirable option in scenarios where the quality (factuality or informativeness or both) of the model response is paramount, or when the inference computation is optimized with parallelism or getting much cheaper (as with the current trend).
>
> **Time Latency** The current implementation can be relatively time-consuming. However, we would like to emphasize that most steps (generating samples, breaking down claims, constructing edges) in our pipeline can be *parallelized* and computed in batch, leveraging GPU capabilities to potentially reduce latency to a great extent.
>
> ### Empirical assumption
>
> > *”Based on my understanding, in the responses from the datasets used in this paper, there are fewer dependencies between claims. Do you consider other long-form generation tasks where there are more complex inter-sentence dependencies?”*
>
> We appreciate this insightful observation and your demonstrating example. While our method implicitly utilizes claim dependencies through closeness centrality, we acknowledge that our initial test datasets may have had fewer inter-claim dependencies. To address this and strengthen the generality of our experiments, we've added the Natural Questions dataset, with results presented in Table 4. One example in the Natural Question is: “When are hops added to the brewing process?” Thus, this dataset probably exhibits more complex inter-sentence relationships.
>
> The results show that our claim-level uncertainty estimate method improves performance in most settings or remains at least comparable to baseline methods.
>
> ### Additional questions & suggestions
>
> > *“Just want to double check, is C_D equal to Self-consistency?”*
> Yes, C_D is indeed equivalent to self-consistency.
>
> > *”The following related work is missing…”*
>
> We appreciate you bringing this to our attention. The work you mentioned is indeed relevant. We commit to including a more comprehensive review of related literature, in the updated version of our paper.
>
> > *”Another suggestion for future work is that the authors could conduct experiments about how these estimated uncertainties benefit users when they are prompting LLMs.”*
>
> This is an excellent suggestion for future work. The impact of estimated uncertainties on user interactions with LLMs is indeed an under-explored area. We agree that this direction holds significant potential and plan to discuss it in the Future Work section of the arXiv version of our paper. Such research could provide valuable insights into improving user experience and the practical applications of uncertainty estimation in LLM interactions.

---

### Official Review · Reviewer_XiBs · 2024-07-09

**Soundness:** 2
**Presentation:** 2
**Contribution:** 2
**Rating:** 5
**Confidence:** 4

**Summary:**

To achieve accurate and fine-grained uncertainty estimation for long-form generation of large language models (LLMs). the authors propose a new framework to measure the uncertainty by building a bipartite graph between the randomly sampled responses from LLMs and the set of the claims within these responses first and then utilizing the existing graph centrality metrics to calculate the uncertainties of each claim. Moreover, the uncertainty metric can further be integrated into the decoding phase of the LLM generation to enhance the factuality of the outputs. Experimental results on two datasets with multiple LLMs show that the proposed method can improve the performance of uncertainty estimation than the baselines with the closeness metric. And the decoding with this metric can achieve a better trade-off between factuality and informativeness.

**Strengths:**

The strengths of the paper are listed as follows.
1. Improving the factuality of the long-form generation is important for the wide application of LLMs and existing methods can still not overcome this challenge with satisfactory performance.
2. The proposed method is clearly illustrated in Figure 1.
3. The proposed method is reasonable. This pipeline can indeed be performed under black-box settings and with fine-grained (i.e., claim level).

**Weaknesses:**

The weaknesses of the paper are listed as follows.
1. The whole pipeline relies heavily on the LLM for many critical steps such as claim decomposition, causal inference and merging semantically duplicated claims, etc. It can thus cause two main concerns. First, if the LLM is not reliable in any of these steps, then the performance will be significantly degraded. Second, the computational cost is heavy due to the frequent use of LLMs.
2. According to the experiments, closeness is the most effective metric for uncertainty estimation. Therefore, it would be better if the authors could give more detailed explanations with examples to calculate it in the proposed framework in the section of method.
3. In the section of related work, the subsection **Granular Uncertainty Estimation** is one type of **Uncertainty Estimation for LLMs**. Therefore, instead of introducing it in a separate subsection, it should be introduced within the section of **Uncertainty Estimation for LLMs**.
4. It would be better if the authors could evaluate their method on more datasets or at least more samples. Randomly sampling 50 entities is not very convincing for evaluation.

**Questions:**

The questions of the paper are listed as follows.
1. Is this work the first one to formulate the uncertainty estimation as a graph-based problem?  What is the main difference between the pipeline of this work and that of the self-consistency in addition to utilizing different graph centrality metrics?
2. For uncertainty-aware decoding, is the whole workflow different from the existing works except for the source of the uncertainty metric?
3. As shown in Figure 4 (b), does it mean that the performance of self-consistency will not change when increasing the number of claim nodes?

**Limitations:**

The authors claim their contributions to the long-form generation. But the claim decomposition is the only step designed specifically for long-form generation. The relationship between the different claims within a single long-form generation is not considered.

---

> ### Author Rebuttal · Authors · 2024-08-07
>
> Thank you for your valuable feedback and suggestions. We appreciate that you acknowledged that our method approaches an important problem and offers satisfactory results. We would like to address your remaining questions and concerns in the following response. **Due to the space limitation of the rebuttal section, we've moved responses to some less critical questions to the official comments. Please review both the following response and the accompanying official comments.**
>
> ### Results on additional datasets and with more samples
> > *”It would be better if the authors could evaluate their method on more datasets or at least more samples. Randomly sampling 50 entities is not very convincing for evaluation.”*
>
> **Additional Dataset**
> Thank you for pointing this out. We acknowledge that more datasets could strengthen our claims. Our initial choice of datasets was constrained by two main challenges:
> - The need for datasets challenging enough for state-of-the-art LLMs to ensure a meaningful balance between true and false claims, making uncertainty metrics more informative.
> - The difficulty in constructing accurate auto-annotation pipelines for claims. Given that ~6000 claims involved across six configurations, manual annotation of all claims is impractical.
>
> To address this concern and strengthen the generality of our experiments, we have added the Natural Questions dataset, with results presented in Table 5. The findings show improvements using closeness centrality in most settings.
>
> **More Samples**
>
> In fact, in our original study with 50 entities (approximately 1000 claims in each configuration), the results are already statistically significant. However, recognizing that 50 entities might not be intuitively convincing, we have now doubled our sample size to 100 entities (Table 3). Our method continues to outperform baselines across both datasets for every model, further strengthening our findings.
>
> ### Computational cost
> > *”The computational cost is heavy due to the frequent use of LLMs.”*
>
> Thank you for highlighting this crucial point about computational cost. We agree that the computation overhead is the major limitation of our method. However, we would like to emphasize that our method offers a desirable option for utilizing more inference-time computation to improve the quality of LLMs’ responses. More importantly, our new results demonstrate that ***our method achieves the Pareto optimality for the compute-quality tradeoff*** compared to existing methods.
>
> **Pareto optimality** To evaluate our method's efficiency, we calculated the compute cost and quality metrics  (factuality, informativeness, and the combined metric ‘ factuality x informativeness’ to balance the two) against the baselines. We conducted extensive hyperparameter tuning for all methods, plotting frontier curves for various configurations. We exclude (SC + IL-VC, Multi-Sample) and (SC, Multi-Sample) since they share the same computational cost as ours due to the similar graph construction, but performs worse than our methods (Figure 3). Our analysis reveals that our method consistently outperforms baseline approaches across all metrics, achieving Pareto optimality. This means that to achieve the same level of output quality, our method is relatively computational efficient.
>
> ### Potential generalization of the proposed method
>
> > *”But the claim decomposition is the only step designed specifically for long-form generation. The relationship between the different claims within a single long-form generation is not considered.”*
>
> We would like to clarify that our method incorporates the inter-claim relationship with graph metrics. A crucial step is the graph propagation (graph-metric calculation), which allows us to consider inter-claim relationships more comprehensively than previous methods.
>
> Consider this example:
> If Claims A and B are supported by an equal number of generations, but A's co-occurring claims are more corroborated across the graph:
> - Self-consistency (degree centrality) would assign equal scores to A and B.
> - Our method would score A higher, as its two-hop neighbors are more strongly connected.
>
> Highly supported claims may indicate greater reliability for co-occurring claims. By considering these broader relationships, our method provides a more nuanced uncertainty measure.
>
> ### Additional questions & suggestions
>
> > *”As shown in Figure 4 (b), does it mean that the performance of self-consistency will not change when increasing the number of claim nodes?”*
>
> Thank you for highlighting this point. We should have clarified it better. In our study, we estimate the uncertainty on a fixed pool of claims entailed by the first generation (as our dataset) to ensure the consistency of measures across different setups, and vary the size of the entire claim set in the graph by using different numbers of sampled responses.
>
> Since self-consistency does not utilize information between claims (such as 2-hop or n-hop relationships in the graph), fixing the number of generations while increasing the number of claims in the graph does not influence the uncertainty estimation performance for the pool of measured claims. In contrast, our method's performance improves, demonstrating that closeness centrality effectively utilizes claim-claim relationship information.

---

> ### Author Response · Authors · 2024-08-07
> **Continuation of the Rebuttal**
>
> ### Reliance on LLMs
> > *”The whole pipeline relies heavily on the LLM for many critical steps… if the LLM is not reliable in any of these steps, then the performance will be significantly degraded.”*
>
> We acknowledge this in the Limitation section. However, this concern would be less critical as LLMs are rapidly becoming more capable. Most of the current top-performing LLMs, including open-source models, are undoubtedly able to perform these tasks reliably. This issue is expected to further reduce in the future.
>
> ### Additional questions & suggestions
>
> > *”Therefore, it would be better if the authors could give more detailed explanations with examples to calculate it in the proposed framework in the section of method.”*
>
> Thank you for this suggestion. We agree that a detailed example would enhance the clarity of our method and will incorporate a more detailed explanation in the Method section of our revised paper.
>
> > *”Is this work the first one to formulate the uncertainty estimation as a graph-based problem? What is the main difference between the pipeline of this work and that of self-consistency in addition to utilizing different graph centrality metrics?”*
>
> We are indeed the first to utilize the graph information of the relationship between generations and claims. While previous work extends semantic entropy to soft clustering using similarity between different generations [1], our approach differs significantly in graph construction (the whole procedure is different), the graph information utilized (graph information in semantic entailment graph), and the task focus (claim-wise uncertainty estimation). Using graphs is generally a common approach when relationships are involved; thus, the key contributions lie in how the graph is constructed and the specific relationship information it captures. Our work identifies the method to construct the graph and extract uncertainty-related information to improve methods on a general new problem – claim-wise uncertainty estimation.
>
> [1] Generating with Confidence: Uncertainty Quantification for Black-box Large Language Models. TMLR 2024.
>
> > *”For uncertainty-aware decoding, is the whole workflow different from the existing works except for the source of the uncertainty metric?”*
>
> The applicability of a wide range of uncertainty metrics is the major difference of our work compared to existing works, a result of our framework's generality that allows for the plug-and-play integration of various claim-wise uncertainty measures. Closest to our work, [2, 3] is only applicable to self-consistency measures. Additionally, [3] differs in its approach by using claims from a single response (despite sampling multiple responses), trading off informativeness. We will elaborate on these distinctions in our related work section.
>
> [2] Wang et al. Fine-grained self-endorsement improves factuality and reasoning. Findings of ACL 2024.
>
> [3] Mohri and Hashimoto. Language models with conformal factuality guarantees. ICML 2024.

---

> > ### Comment · Reviewer_XiBs · 2024-08-14
> >
> > Thanks for the detailed feedback from the authors. The response addresses some of my concerns. However, the issue of the reliance on the performance of LLM in multiple steps is still not addressed. Although the authors claim that applying more capable LLMs can mitegate this issue, it is highly likely to introduce more computational costs. And intensive computational cost is already an issue of this method. Moreover, the main technical contribution is moderate. The main difference from existing works is the different sources of the uncertainty metric. Besides, this work does not consider the relationship between the different claims within a single long-form generation, which is listed by me in the section of limitations. There, I prefer to maintain my original ratings.

---

> ### Author Response · Authors · 2024-08-14
> **Express gratitude to reviewer's comment and further clarification**
>
> Thank you for your response! We appreciate your feedback and understand your decision to maintain the original rating. However, we believe there may be some misunderstandings that we'd like to address:
>
> **Computational Efficiency**: It makes sense that future models might get larger but the trend of inference cost is becoming cheaper in general, these subtasks might be completed more efficiently in the future.
>
> **Inter-Claim Dependency**: A key advantage of our method is its utilization of claim dependencies within long-form generations, which we should have emphasized more clearly in our initial response. In the example that we provided:
>
> > Consider this example: If Claims A and B are supported by an equal number of generations, but A's co-occurring claims are more corroborated across the graph:
> > - Self-consistency (degree centrality) would assign equal scores to A and B.
> > - Our method would score A higher, as its two-hop neighbors are more strongly connected.
>
> Claim A will be assigned a less uncertain score than claim B because the other claims in the same long-form generation are more corroborated, even if they appear in the same number of generations. This example conveys a general idea of utilizing "closeness" on the semantic entailment graph: Claims that are less uncertain will result in less uncertain scores in claims from the same generation, which considers the correlation between uncertainty scores between claims within a long-form generation. In terms of the uncertainty estimation method, this also demonstrates where our method differs from previous methods: not only the graph metric, but also the idea that we should utilize the uncertainty from other claims (in both the same and different generations). To our knowledge, no prior uncertainty method has attempted to break down claims from multiple generations and exploit inter-claim relationships in this manner.
>
> We hope this clarification addresses your concerns and better explains our work. We appreciate your time and consideration.

---

### Official Review · Reviewer_29kR · 2024-07-13

**Soundness:** 3
**Presentation:** 3
**Contribution:** 3
**Rating:** 5
**Confidence:** 4

**Summary:**

This paper introduces Graph Uncertainty, a novel framework for estimating granular, claim-level uncertainty in long-form LLM outputs using semantic entailment graphs and graph centrality metrics. The authors claim the proposed method can go beyond existing works that can only be applied to multiple-choice questions or their entire generated response. The proposed methods show improvements in claim-level uncertainty estimation and factuality of generated responses. Overall, the paper is easy to follow and comprehensive.

**Strengths:**

1. The introduction of graph-based uncertainty metrics for claim-level analysis is novel and addresses a critical gap in LLM uncertainty estimation.
2. The use of various graph centrality metrics to assess claim importance and uncertainty is interesting and systematically explored.
3. The integration of uncertainty estimates into decoding processes shows practical applications, improving the reliability of LLM outputs.

**Weaknesses:**

1. The related work section (and possible experiment comparison) lacks some important works. In addition, although various methods are benchmarked, the paper could benefit from additional comparisons with state-of-the-art uncertainty estimation techniques not limited to graph-based methods.
2. The proposed method's graph construction process is computationally intensive and may not be feasible for all real-world applications.
3. The framework assumes claims are largely independent, which might not hold in all real-world scenarios.

**Questions:**

1. The related work section (and possible experiment comparison) lacks some important works [1,2] that leverage the entropy-based framework to decompose the uncertainty of LLM response.
2. It seems like graph-based uncertainty is another form of variance-based uncertainty estimation.
3. The whole concept of claim extraction seems to create extra work. For some QA tasks, we only need a direct answer, not all the associated replies. Creating the graph would bring lots of computational costs.
4. The paper primarily uses FactScore and PopQA datasets; more diverse datasets could strengthen the generalizability of the results.


[1] Ling, Chen, et al., "Uncertainty Quantification for In-Context Learning of Large Language Models." (NAACL 2024)
[2] Fadeeva, Ekaterina, et al. "LM-polygraph: Uncertainty estimation for language models." (EMNLP 2023)

---

> ### Author Rebuttal · Authors · 2024-08-07
>
> Thank you for your valuable feedback and suggestions. We appreciate that you acknowledge our work is “novel and addresses a critical gap in LLM uncertainty estimation”. We would like to address your remaining questions and concerns in the following response.
>
> ### Computational cost
> > *”The proposed method's graph construction process is computationally intensive and may not be feasible for all real-world applications.”*
>
> We agree that the computation overhead is the major limitation of our method. However, we would like to emphasize that our method offers a desirable option for utilizing more inference-time compute to improve the quality of LLMs’ responses. More importantly, our new results demonstrate that ***our method archives the Pareto optimality for the compute-quality tradeoff*** compared to existing methods.
>
> **Pareto optimality** We calculated the compute cost and various quality metrics (including  ‘factuality’, ‘informativeness’, and the combined metric ‘ factuality x informativeness’ ) of our method and baseline method. We conducted a comprehensive analysis by tuning hyperparameters and plot the frontier curves for each method. We do not include (SC + IL-VC, Multi-Sample) and (SC, Multi-Sample) since they share the same computational cost as ours due to a same graph computation procedure, but their performance is worse than our method (Figure 3). We observe that *our method achieves Pareto optimality in all the metrics.
>
> **Real-world applicability** In real-world applications, time latency is often a more critical consideration than raw computation at inference time. We would like to emphasize that most steps (generating, breakdown, constructing edges) in our pipeline can be *parallelized* and computed in batch, leveraging GPU capabilities to potentially reduce latency to a great extent.
>
> > *”The whole concept of claim extraction seems to create extra work. For some QA tasks, we only need a direct answer, not all the associated replies. Creating the graph would bring lots of computational costs.”*
>
> We acknowledge that a direct answer suffices in many scenario. However, in cases where factuality and informativeness are critical, the quality-cost trade-off becomes relevant. Our method provides an option for users to choose when the additional computational investment is warranted by the need for improved output quality.
> ### Additional datasets
> > *“The paper primarily uses FactScore and PopQA datasets; more diverse datasets could strengthen the generalizability of the results.”*
>
> Thank you for highlighting this important point. We acknowledge that more datasets could strengthen our claims. Our choice of datasets was constrained by two main challenges:
> - The need for datasets challenging enough for state-of-the-art LLMs to ensure a meaningful balance between true and false claims, making uncertainty metrics more informative.
> - The difficulty in constructing accurate auto-annotation pipelines for claims. Given that our previous experiments involved ~6000 claims across six configurations, manual annotation of all claims is impractical.
>
> To strengthen the generality of our experiments, we have added the Natural Questions dataset, with results in Table 4. The findings show improvements using closeness centrality in most settings.
>
> ### Existing work and comparison
> > *”The related work section (and possible experiment comparison) lacks some important works. In addition, although various methods are benchmarked, the paper could benefit from additional comparisons with state-of-the-art uncertainty estimation techniques not limited to graph-based methods.”*
>
> We appreciate this observation. It's important to clarify that our comparison is not limited to graph-based methods. Our aim is to compare various black-box LLM uncertainty estimation methods, aligning with the general problem setting in our paper. This focus on black-box methods excludes many logit-based techniques, which are not applicable in this context.
>
> > *”The related work section (and possible experiment comparison) lacks some important works [1,2] that leverage the entropy-based framework to decompose the uncertainty of LLM response.”*
>
> We appreciate your highlighting these important works. Due to space constraints, we had to condense the Related Work section in the current version. However, we commit to including a more comprehensive review of relevant literature, including the entropy-based framework approaches you mentioned, in the updated version of our paper.
>
> ### Empirical assumption
>
> > *“The framework assumes claims are largely independent, which might not hold in all real-world scenarios.”*
>
> We believe you're referring to the claim breakdown process. It is true that this step assumes claims are separable, which may overlook strong dependencies between claims. However, our uncertainty estimation method actually better considers claim dependencies than previous approaches when measuring uncertainty.
>
> Our approach utilizes broader graph information, including connections beyond immediate neighbors. This allows us to capture more nuanced relationships between claims. For example:
>
> Consider two claims, A and B, each supported by an equal number of generations. In a self-consistency approach, these would receive identical scores. However, our method can differentiate between them based on the broader context:
>
> - If the other claims in generations supporting A are generally more corroborated across the graph than those supporting B, our closeness centrality metric would assign a higher score to A. This is because A's two-hop neighbors (other claims in the same generation) are more connected to other nodes in the graph.
>
> When highly supported claims appear in a generation, they may indicate a greater likelihood of reliability for other claims in the same context. Closeness centrality implicitly leverages the dependencies between claims within a generation, offering a more fine-grained approach to uncertainty estimation.

---

### Author Rebuttal · Authors · 2024-08-07

We thank all reviewers for their helpful feedback and suggestions.

We are glad that the reviewers found our work our method approaches an important problem [29kR], proposes a novel and interesting method [29kR, 2gkm], offers thorough empirical results [29kR, 2gkm], and “addresses a critical gap in LLM uncertainty estimation” [29kR].

We have provided extensive empirical results (included in the supplementary PDF) and responses to address reviewers’ remaining questions. Specifically, we have attempted to address the following major ones:

[Reviewer 29kR, 29kR, 2gkm] Computational cost: Even though our method is indeed introducing additional inference for constructing the graph, our empirical experiments show that our method achieves the **Pareto optimality for the compute-quality tradeoff** compared to other existing methods. These results are illustrated in Figure 5.

[Reviewer 29kR, XiBs] Results on additional datasets and increased sample size: We have enhanced the main table of uncertainty estimation experiments by **doubling the data size** (100 data points with 1700-2400 claims in each experimental setting) for improved statistical power. Furthermore, we have included experiments on **an additional dataset**, Natural Questions, to demonstrate better generalizability. These results are presented in Tables 3 and 4.

[Reviewer 29kR, 2gkm] Empirical assumption of our method: We demonstrate that even though our breakdown process assumes that claims are separable, a key advantage of our method is its implicit utilization of claim dependencies within each graph. This is achieved by leveraging information beyond direct neighbors (degree centrality). The details are discussed in each individual response.

Please see detailed responses to specific questions below.

---

### Decision · Program_Chairs · 2024-09-25

**Decision:**

Accept (spotlight)

**Comment:**

The paper introduces a novel method for estimating claim uncertainty in long-form LLM generation using semantic entailment graphs and graph centrality metrics. They also introduce a new decoding approach for long-form generation.  Both methods demonstrate improved performance both in uncertainty estimation and in more accurate generation.  Evaluation is thorough and clear.  Paper is well-written.
The only remaining weakness after the rebuttal/discussion period is computational cost of the method.